# Instance-aware Generalized Multi-task Visual Grounding

## Abstract

The recently proposed Generalized Referring Expression Segmentation (GRES) and Comprehension (GREC) tasks extend the traditional RES/REC paradigm by incorporating multi-target and non-target scenarios. However, the existing approaches focus on these tasks individually, leaving the unified generalized multi-task visual grounding unexplored. Moreover, current GRES methods are limited to global segmentation, lacking fine-grained instance-level awareness. To address these gaps, this paper introduces a novel **I**nstance-aware **G**eneralized multi-task **V**isual **G**rounding (**IGVG**) framework. IGVG is the first to integrate GREC and GRES, establishing a consistent correspondence between detection and segmentation via query guidance. Additionally, IGVG introduces instance-level awareness, enabling precise and fine-grained instance recognition. Furthermore, we present a Point-guided Instance-aware Perception Head (PIPH), which employs attention-based query generation to identify coarse reference points. These points guide the correspondence between queries, objects, and instances, enhancing the directivity and interpretability of the queries. Experimental results on the gRefCOCO (GREC/GRES), Ref-ZOM, and R-RefCOCO/+/g benchmarks demonstrate that IGVG outperforms state-of-the-art methods.

## 1 Introduction

Classic Referring Expression Perception (REP) tasks primarily include Referring Expression Comprehension (REC) (Yu et al., 2018a; Yang et al., 2020; Shi et al., 2023) and Referring Expression Segmentation (RES) (Liu et al., 2023e; Shang et al., 2024; Chen et al., 2024). The goal of REC is to locate a specific target based on textual descriptions, while RES aims to achieve more fine-grained, pixel-level localization. Both REC and RES share a common characteristic, *i.e.*, one-to-one correspondence between textual description and target object. Recently, the generalized REP tasks, such as Generalized REC (GREC) (He et al., 2023) and RES (GRES) (Liu et al., 2023a), have been proposed to extend the applicability of classic methods by involving multiple/non-target scenarios.

Furthermore, numerous studies have focused on multi-task learning, which aims to tackle detection and segmentation with a unified architecture. MCN (Luo et al., 2020) is one of the first approaches to combine REC and RES while investigating consistency constraints. Subsequent works (Li & Sigal, 2021; Zhu et al., 2022; Liu et al., 2023d) have primarily concentrated on harnessing the complementary strengths across multiple tasks. However, the effectiveness of joint multi-task learning in generalized scenarios has yet to be explored and validated. In this paper, we heuristically construct a *generalized multi-task* visual grounding framework that simplifies task complexity while achieving complementary predictions. As illustrated in Fig. 1(c), we adopt a straightforward architecture in which two independent decoders for detection and segmentation are jointly trained to build a bridge between GREC and GRES tasks.

As illustrated in Fig. 1(b), most existing GRES methods (Liu et al., 2023a; Zhang et al., 2024; Xia et al., 2024; Luo et al., 2024) merge all target masks into a single global mask, effectively treating the task as semantic segmentation that classifies all targets as a unified foreground against the background. These methods exhibit certain limitations. On the one hand, the models lack instance-level perception during training, resulting in the loss of fine-grained supervision. On the other hand, their predictions fail to capture instance-level awareness, yielding only coarse foreground masks, which are inadequate for scenarios requiring detailed instance perception. In this paper, we

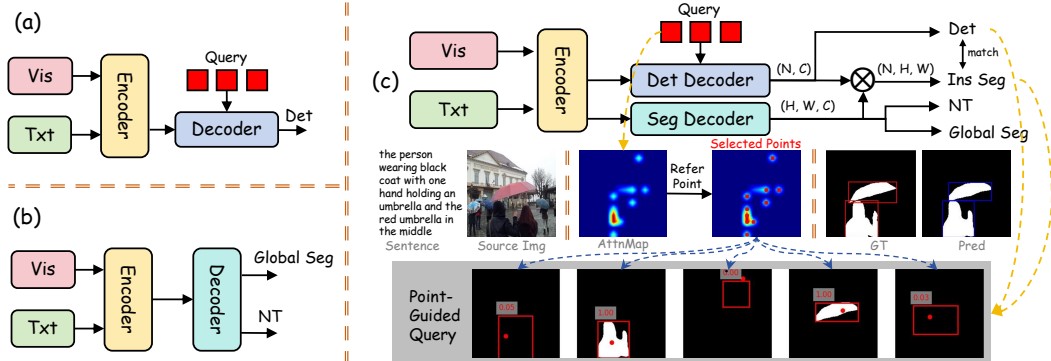

Figure 1: **Comparison of Different Tasks:** (a) The transformer-based GREC paradigm; (b) Common GRES tasks that predict the global mask and non-target branch separately; (c) The proposed IGVG framework, which integrates GREC and GRES to enable instance-level referring segmentation. Additionally, instance-aware queries are guided and constructed through adaptive selection of reference points. We visualize the boxes and masks predicted by five queries.

combine both *instance-level* and global semantic supervision to equip the model with instance-aware capabilities, while enhancing perceptual performance through multi-granularity joint learning of referred targets. Additionally, by establishing associations between queries, objects, and instances, we ensure consistent predictions for both bounding boxes and masks from the same query.

Moreover, existing GREC methods can be illustrated in Fig. 1(a), which follows a query-based architecture. Many recent studies focus on query design, including the incorporation of prior information (Meng et al., 2021; Liu et al., 2022) and the exploration of query matching strategies (Li et al., 2022a; Zhang et al., 2022; Chen et al., 2023). These methods employ handcrafted or learnable priors to enhance the directivity of queries, thereby accelerating convergence during training. However, in the context of the GREC task, the text typically references regional image content rather than the whole. Therefore, queries can be generated following regional spatial priors regarding the text references. In this paper, we propose a *point-guided* instance-aware perception head that adaptively selects prior points and utilizes them to identify corresponding instance locations, thereby enhancing the interpretability of query predictions. As shown in Fig. 1(c), we filter prior reference points based on attention distributions and direct the queries toward the nearest corresponding targets through point-guided target matching. The last row of Fig. 1(c) illustrates the predicted boxes and masks for five queries alongside their prior points.

The main contributions of this paper are summarized as follows:

- We propose an Instance-aware Generalized multi-task Visual Grounding (IGVG) framework, which pioneeringly addresses the GREC and GRES tasks simultaneously.

- IGVG possesses instance-aware segmentation capabilities while ensuring consistent predictions with the detection task. Additionally, it integrates global semantic segmentation to achieve multi-granularity predictions, further enhancing robustness.

- We develop a Point-guided Instance-aware Perception Head (PIPH) that improves the directivity and interpretability of query-to-target correspondence.

- The proposed IGVG framework achieves promising results on the gRefCOCO (GREC), gRefCOCO (GRES), Ref-ZOM, and R-RefCOCO/+/g datasets, significantly outperforming the state-of-the-art approaches.

## 2 RELATED WORK

**Traditional Referring Expression Comprehension/Segmentation.** In conventional REC, one sentence corresponds to a single target bounding box. Early two-stage methods (Yu et al., 2018a;b; Hong et al., 2019; Liu et al., 2019; Chen et al., 2021) tackled this problem by first generating candidate proposals and then matching the referring expression to the proposals. Later, one-stage methods (Zhou et al., 2021; Luo et al., 2020; Yang et al., 2020) adopted a dense anchor strategy to enable

efficient inference. In recent years, many Transformer-based methods (Ye et al., 2022; Zhu et al., 2022; Deng et al., 2021; Su et al., 2023a) have been proposed to effectively capture cross-modal relationships. RES, on the other hand, is a task where one sentence corresponds to a set of pixels. Classical approaches (Hu et al., 2016; Liu et al., 2017; Huang et al., 2020; Feng et al., 2021) typically rely on convolution-based operations for cross-modal fusion to generate segmentation masks. To address the limitation of insufficient visual-language relation modeling in previous works, recent studies (Yang et al., 2022; Ding et al., 2021; Liu et al., 2023b;e; Kim et al., 2022) have employed advanced attention-based mechanisms (Vaswani et al., 2017) to enhance the multimodal interaction. In particular, SimVG (Dai et al., 2024) improves referential understanding by decoupling multimodal fusion from downstream tasks to upstream pre-training. Building on this, our work focuses on generalized scenarios and introduces an adaptively point-guided multi-task visual grounding approach.

**Generalized Referring Expression Comprehension/Segmentation.** Recently, to mitigate the inflexibility of REC with one-to-one pairing, ReLA (Liu et al., 2023a) introduced the GRES task, which expanded the scope to include both empty-target and multiple-target scenarios. Furthermore, GREC (He et al., 2023) extended GRES from segmentation to detection tasks. Similarly, DMMI (Hu et al., 2023) introduced a new benchmark and baseline for beyond-single-target segmentation, and RefSegformer (Wu et al., 2024) equipped transformer-based models with empty-target sentence discrimination, achieving robust segmentation performance. Nonetheless, all these methods predict a global mask that combines all instances, directly overlooking the importance of fine-grained instance-level information. In contrast, the proposed approach reuses instance annotations by incorporating instance-level in addition to global semantics supervision. This not only enhances the model's instance-aware capability but also ensures consistent predictions between target and instances.

**Multi-Task Visual Grounding (MTVG)**. MTVG aims to localize and segment referring expressions using a single integrated model. Some Transformer-based methods (Luo et al., 2020; Li & Sigal, 2021; Su et al., 2023b; Chen et al., 2024) have pursued more comprehensive multimodal modeling approaches to improve the performance of multi-task visual grounding. SeqTR (Zhu et al., 2022) and PolyFormer (Liu et al., 2023d) utilized a sequential transformer model that processes visual and textual data in a unified manner, sequentially refining predictions to enhance multi-task visual grounding performance. Recently, LLM-based methods (Peng et al., 2023; Lai et al., 2024; Xia et al., 2024; Rasheed et al., 2024) harnessed the capabilities of large language models (LLMs) to enforce rule-based serialization of predictions, effectively integrating the REC and RIS tasks within a unified framework. However, multi-task visual grounding in generalized scenarios remains underexplored. To bridge this gap, this paper pioneeringly presents a solution that combines GREC and GRES in a single framework.

# 3 THE PROPOSED IGVG METHOD

In this section, we provide an overview of the IGVG architecture, as illustrated in Fig. 2. The process begins by independently embedding and processing an image $\mathcal{I} \in \mathbb{R}^{H \times W \times 3}$ and a textual expression $\mathcal{E}$ using a Multi-Modality Encoder (MME) (Wang et al., 2023), which performs vision-language encoding and fusion. The MME outputs include visual features $\mathcal{F}_i' \in \mathbb{R}^{N_i \times C'}$ and textual features $\mathcal{F}_t' \in \mathbb{R}^{N_t \times C'}$. Next, we respectively apply image projection (IP) and text projection (TP), to map $\mathcal{F}_i'$ and $\mathcal{F}_t'$ to a lower dimension $C$, yielding $\mathcal{F}_i$ and $\mathcal{F}_t$. The architecture then branches into two core components. The first component is the proposed Point-guided Instance-aware Perception Head (PIPH), depicted in the light blue section of Fig. 2. Its primary function is to adaptively filter key prior points, match them with the corresponding targets, and perform both object detection and instance-level segmentation. This part will be elaborated in Sec. 3.1. The second component inherits the global segmentation approach of the mainstream GRES methods. It first utilizes SimFPN (Li et al., 2022b) to extend the single-layer output of ViT to multi-scale features. A simple Unet decoder is then employed to fuse hierarchical information and produce global segmentation results $S_{global} \in \mathbb{R}^{H' \times W' \times C}$. This result serves three purposes: 1) interacting with object queries to generate instance-aware semantic queries for instance segmentation; 2) providing global segmentation predictions; 3) guiding non-target predictions. It is important to note that this section focuses primarily on the core innovations of our method. Some details, such as the STS and post-processing steps, are provided in Appendix D.

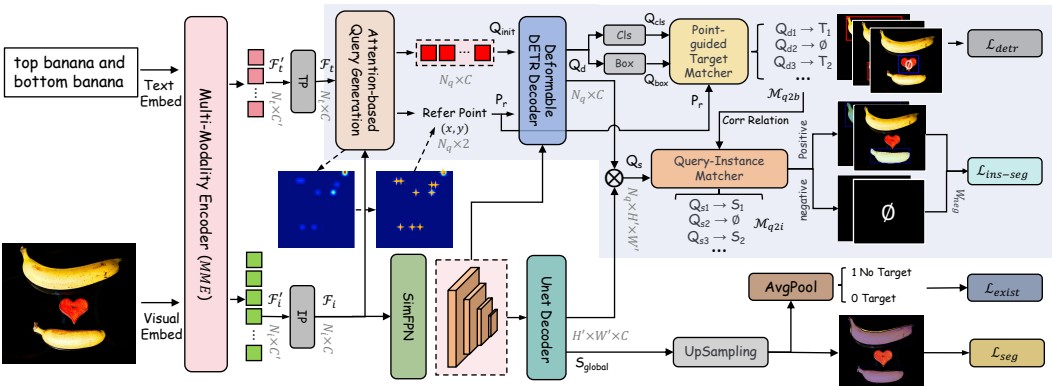

Figure 2: **Overview of IGVG**. The Multi-Modality Encoder (MME) simultaneously fuses the referring expression and image. After obtaining the fused features, $\mathcal{F}_t$ and $\mathcal{F}_i$, one branch performs global semantic segmentation and non-target prediction. The other branch highlighted in light blue is the proposed Point-guided Instance-aware Perception Head (PIPH) which filters reference points using an attention map. These points are then used as initial positions in the deformable decoder, guiding the final target and instance predictions.

### 3.1 THE POINT-GUIDED INSTANCE-AWARE PERCEPTION HEAD (PIPH)

As shown in the light blue part of Fig. 2, the core idea of PIPH is to adaptively filter prior reference points through the Attention-guided Query Generation module (AQG), guiding subsequent object detection and instance segmentation. This not only enhances the interpretability of the queries but also achieves finer-grained perception. Specifically, an attention-based query selector is first used to select the query set $Q_{init} \in \mathbb{R}^{N_q \times C}$ and the initial prior point set $P_r \in \mathbb{R}^{N_q \times 2}$. These are then passed through a deformable DETR decoder (Zhu et al., 2020) to dynamically query contextual information from multi-scale image features, generating refined queries $Q_d \in \mathbb{R}^{N_q \times C}$ that capture different target contextual semantics.

Subsequently, the model computes the cost between all targets through the prior points, predicted boxes, and confidence scores. The optimal assignment between queries and targets $\mathcal{M}_{q2b}$ is determined using the Hungarian algorithm (Carion et al., 2020). On the one hand, the model directly computes the DETR object detection loss. On the other hand, $Q_d$ is multiplied with the global segmentation result $S_{global}$ to obtain a response mask $Q_s \in \mathbb{R}^{N_q \times H' \times W'}$ for each query within the global context. Using $\mathcal{M}_{q2b}$, we pass this information to the instance mask matching process, constructing the pairing relationships $\mathcal{M}_{q2i}$. This ensures the consistency between predicted boxes and masks for each query with respect to their corresponding targets. Last, the instance segmentation loss, $\mathcal{L}_{ins-seg}$, supervises both the positive and negative sample masks to reinforce accurate instance segmentation.

#### 3.1.1 THE ATTENTION-GUIDED QUERY GENERATION (AQG) MODULE

AQG aims to adaptively select suitable initial reference points and query embeddings. To this end, as shown in Fig. 3, we first use a Score Text Selector (STS) to filter $N_q$ effective queries from the $N_q$ text tokens, which are then used as Q in cross-attention with image patches (K/V). The attention map is obtained by:

$$\mathcal{M}_{\text{attn}} = \text{Softmax}\left(\frac{\mathcal{F}_{\text{filter}} \cdot \mathcal{F}_i^{\top}}{\sqrt{d_k}}\right) \cdot \mathcal{F}_i. \tag{1}$$

Next, we perform average pooling on $\mathcal{M}_{\text{attn}}$ across the $N_q$ channels to obtain the spatial score distribution map $\mathcal{M}_s$. Then, we employ a Dist-Score Point Selector to adaptively select prior location points $P_r$ from $\mathcal{F}_i$ that cover all possible referential instances and their corresponding queries $Q_{filter}$. By concatenating $Q_{filter}$ with $\mathcal{F}_q$ and passing them through an MLP layer, we obtain the initial query embeddings $Q_{init}$. This process can be expressed as follows:

$$\mathcal{M}_s = \text{AvgPool}(\mathcal{M}_{\text{attn}}), \qquad Q_{\text{init}} = \text{MLP}(\text{Concat}(Q_{filter}, \mathcal{F}_q)). \tag{2}$$

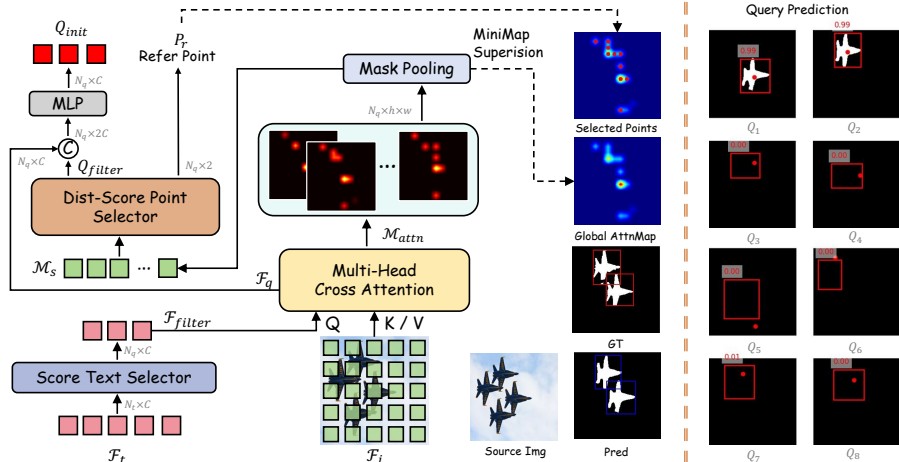

Figure 3: **The AQG module**. It selects initial reference points and contextually rich the initial queries. First, a Score Text Selector (STS) filters a small number of effective and highly responsive tokens from $\mathcal{F}_t$, which are used as query (Q) in the multi-head cross attention, with $\mathcal{F}_i$ serving as key (K) and value (V). Then the Dist-Score Point Selector (DSPS) selects points based on distance and response scores to cover as many referred instances as possible. $Q_1, Q_2, \ldots, Q_8$ are point, box, and mask prediction of 8 queries.

**Score Text Selector (STS)** selects effective and highly responsive tokens from the text token set. This process balances two main factors. First, it excludes padding tokens to retain only the tokens containing meaningful information. Second, it evaluates each token's responsiveness using an L2 norm score. This strategy preferentially selects the tokens with high scores and valid content. The details for STS are described in the Appendix (Algorithm 2).

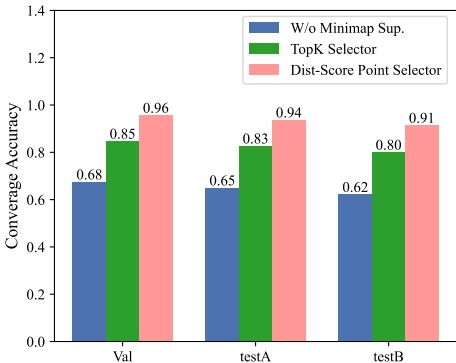

Figure 4: **Bar chart of CoverAcc.** The 'W/o Minimap Sup.' bar represents the results without supervision on the attention map. The 'TopK Selector' bar indicates the use of the TopK strategy to select $N_q$ queries.

---

**Algorithm 1** Dist-Score Point Selector

**Require:** Input attnmap $\mathbf{M} \in \mathbb{R}^{H \times W}$, num of points $N_q$, distance weight $W_{\text{dist}}$
**Ensure:** Selected points $\mathbf{R} \in \mathbb{R}^{N_q}$
1: Apply sigmoid: $\mathbf{M} \leftarrow \sigma(\mathbf{M})$
2: Initialize set $\mathbf{R}$
3: Candidate points: $\mathbf{P} = \{(i,j) \mid i \in [1, H], j \in [1, W]\}$
4: Find max point: $\mathbf{p}_{\max} = \arg\max(\mathbf{M})$
5: Add $\mathbf{p}_{\max}$ to $\mathbf{R}$ and remove from $\mathbf{P}$
6: **for** $k = 1$ to $N_q - 1$ **do**
7:     Compute minimum distance from each point in $\mathbf{P}$ to any point in $\mathbf{R}$
8:     Compute combined score: $\mathbf{S} = \mathbf{M} + W_{\text{dist}} \times \mathbf{D}$
9:     Select best point: $\mathbf{p}_{\text{best}} = \arg\max(\mathbf{S})$
10:     Add $\mathbf{p}_{\text{best}}$ to $\mathbf{R}$ and remove from $\mathbf{P}$
11: **end for**
12: **return** Selected points $\mathbf{R}$

---

**Dist-Score Point Selector (DSPS)** ensures that the selected points do not merely concentrate on a few specific instances. Instead, the selection covers as many potential target instances as possible. DSPS is based on a greedy algorithm, as described in Algorithm 1. The selection criterion for points is that the chosen points should not only have high scores but also be as distant as possible from the previously selected points, corresponding to line 8 in Algorithm 1. Finally, based on the selected set of points $R = \{(x_i, y_i) \mid i = 1, 2, \ldots, N_q\}$, the corresponding queries $Q_{filter}$ are choosen from $\mathcal{F}_i$. To rigorously express the advantages of DSPS, we introduce a metric called Coverage Accuracy (CoverAcc), which measures the quality of points based on their coverage of targets:

$$\text{CoverAcc} = \frac{1}{N} \sum \frac{TP}{TP + FN} \tag{3}$$

where TP denotes the number of targets covered by points within the target box regions, and FN represents the number of targets not covered by any point. The denominator reflects the total number of targets. From Fig. 4, we can draw two conclusions. First, supervision from minimap enhances the reliability of point selection in the attention map. Second, DSPS significantly improves the coverage of instances by reference points.

### 3.1.2 The Instance-aware Point-guided Matcher

The core idea of the matching process is to guide the query-to-target matching relationship through reference points. This process can be divided into two main components: 1) Using reference points to guide the matching between $Q_d$ and targets. 2) Establishing the matching relationship between $Q_s$ and instances based on the correspondence between boxes and instances. This strategy ensures that the target and instance associated with each query remain consistent.

**Point-guided Target Matcher.** Compared to traditional Hungarian matching in DETR, we introduce an additional weighting term in the cost matrix that accounts for the distance between the point and the center of the target box. This modification establishs a more direct association between the initial reference points and the target objects. The cost is defined as:

$$
\begin{aligned}
C_{ij} = {} & \lambda_{\text{cls}} \cdot \text{CE}(p_i^{\text{cls}}, \hat{p}_j^{\text{cls}}) + \lambda_{\text{box}} \cdot \text{L}_1(p_i^{\text{box}}, \hat{p}_j^{\text{box}}) \\
& + \lambda_{\text{giou}} \cdot \text{GIoU}(p_i^{\text{box}}, \hat{p}_j^{\text{box}}) + \lambda_{\text{point}} \cdot \text{L}_1(p_i^{\text{point}}, \hat{p}_j^{\text{center}_{\text{box}}}).
\end{aligned}
\tag{4}
$$

**Query-Instance Matcher.** After applying the point-guided target matcher, we obtain the query-to-object matching relationship $\mathcal{M}_{q2b}$. The query-instance matcher then propagates $\mathcal{M}_{q2b}$ to establish the query-to-instance matching relationship $\mathcal{M}_{q2i}$, given the one-to-one correspondence between objects and instances. In essence, $Q_d$ contains the target's positional information, while $S_{global}$ provides global semantic information. The dot product between $Q_d$ and $S_{global}$ effectively captures the process where queries use feature similarity comparisons to generate instance-aware semantic masks. Thus, $Q_s$ can be interpreted as an instance-level semantic query.

### 3.2 Training Objectives

The training objective includes four components. 1) Detection: The detection branch employs a loss function $\mathcal{L}_{detr}$ similar to DETR, incorporating the L1, Cross-Entropy, and GIoU loss functions to handle the detection task. 2) Global Segmentation: This branch uses the BCE and Dice loss functions, similar to those used in (Liu et al., 2023e), to quantify the difference between the ground truth and predicted global masks: $\mathcal{M}_{gt}$ and $S_{global}$. 3) Instance Segmentation: The instance segmentation branch also utilizes the same BCE and Dice losses as the global segmentation branch, but applies additional weighting to balance positive and negative samples. 4) The Non-Target Branch: This branch is responsible for binary classification and employs the BCE loss to distinguish between target and non-target regions. The total loss function is defined as:

$$
\mathcal{L}_{total} = \lambda_{grec} \cdot \mathcal{L}_{detr} + \lambda_{global} \cdot \mathcal{L}_{seg} + \lambda_{instance} \cdot \mathcal{L}_{ins-seg} + \lambda_{exist} \cdot \mathcal{L}_{exist},
\tag{5}
$$

where the instance segmentation loss $\mathcal{L}_{ins-seg}$ is computed as:

$$
\mathcal{L}_{ins-seg} = \frac{1}{N_{pos}} \sum_i \mathcal{L}_{seg}^{(pos)} + \frac{\lambda_{neg}}{N_{neg}} \sum_j \mathcal{L}_{seg}^{(neg)},
\tag{6}
$$

where positive and negative sample weights are accounted for in the instance segmentation. The default settings for hyperparameters can be found in Appendix C.

## 4 Experimental Results

### 4.1 Experiment Setups

We evaluate our model on four benchmarks, gRefCOCO (GREC/GRES), R-RefCOCO/+/g, and Ref-ZOM, with the official evaluation metrics. Each benchmark and metrics are elaborated in Appendix A and Appendix B. Limited to the space, the implementation details are described in Appendix C, and more ablation studies are referred to Appendix E.

| Method | Backbone | Val | | | TestA | | | TestB | | |
|---|---|---|---|---|---|---|---|---|---|---|
| | | gIoU | cIoU | N-acc. | gIoU | cIoU | N-acc. | gIoU | cIoU | N-acc. |
| *MLLM Methods* | | | | | | | | | | |
| LISA-V-7B (Lai et al., 2024) (ft) | SAM-ViT-H | 61.63 | 61.76 | 54.67 | 66.27 | 68.50 | 50.01 | 58.84 | 60.63 | 51.91 |
| GSVA-V-7B (Xia et al., 2024) (ft) | SAM-ViT-H | 66.47 | 63.29 | 62.43 | 71.08 | 69.93 | 65.31 | 62.23 | 60.47 | 60.56 |
| *Specialist Methods* | | | | | | | | | | |
| MattNet (Yu et al., 2018a) | ResNet-101 | 48.24 | 47.51 | 41.15 | 59.30 | 58.66 | 44.04 | 46.14 | 45.33 | 41.32 |
| LTS (Jing et al., 2021) | DarkNet-53 | 52.70 | 52.30 | - | 62.64 | 61.87 | - | 50.42 | 49.96 | - |
| VLT (Ding et al., 2021) | DarkNet-53 | 52.00 | 52.51 | 47.17 | 63.20 | 62.19 | 48.74 | 50.88 | 50.52 | 47.82 |
| CRIS (Liu et al., 2023e) | CLIP-R101 | 56.27 | 55.34 | - | 63.42 | 63.82 | - | 51.79 | 51.04 | - |
| LAVT (Yang et al., 2022) | Swin-B | 58.40 | 57.64 | 49.32 | 65.90 | 65.32 | 49.25 | 55.83 | 55.04 | 48.46 |
| ReLA (Liu et al., 2023a) | Swin-B | 63.60 | 62.42 | 56.37 | 70.03 | 69.26 | 59.02 | 61.02 | 59.88 | 58.40 |
| HDC (Luo et al., 2024) | Swin-B | 68.28 | 65.42 | 63.38 | 72.52 | 71.60 | 65.29 | 63.85 | 62.79 | 60.68 |
| **IGVG (Ours)** | ViT-B | **73.36** | **69.22** | **72.84** | **75.21** | **74.51** | **71.09** | **66.74** | **65.67** | **65.18** |

Table 1: Comparison with the state-of-the-art methods on gRefCOCO. -V-7B means Vicuna-7B. (ft) denotes that the model is finetuned on the training set of gRefCOCO.

| Method | R-RefCOCO | | | R-RefCOCO+ | | | R-RefCOCOg | | |
|---|---|---|---|---|---|---|---|---|---|
| | mIoU | mRR | rIoU | mIoU | mRR | rIoU | mIoU | mRR | rIoU |
| CRIS (Liu et al., 2023e) | 43.58 | 76.62 | 29.01 | 32.13 | 72.67 | 21.42 | 27.82 | 74.47 | 14.60 |
| EFN (Feng et al., 2021) | 58.33 | 64.64 | 32.53 | 37.74 | 77.12 | 24.24 | 32.53 | 75.33 | 19.44 |
| VLT (Ding et al., 2021) | 61.66 | 63.36 | 34.05 | 50.15 | 75.37 | 34.19 | 49.67 | 67.31 | 31.64 |
| LAVT (Yang et al., 2022) | 69.59 | 58.25 | 36.20 | 56.99 | 73.45 | 36.98 | 59.52 | 61.60 | 34.91 |
| LAVT+ (Yang et al., 2022) | 54.70 | 82.39 | 40.11 | 45.99 | 86.35 | 39.71 | 47.22 | 81.45 | 35.46 |
| RefSegformer (Wu et al., 2024) | 68.78 | 73.73 | 46.08 | 55.82 | 81.23 | 42.14 | 54.99 | 71.31 | 37.65 |
| HDC (Luo et al., 2024) | 74.35 | 83.69 | 52.81 | 64.85 | 87.51 | 49.09 | 65.11 | 84.19 | 43.85 |
| **IGVG (Ours)** | **76.73** | **92.15** | **62.41** | **69.73** | **94.63** | **59.13** | **70.16** | **92.30** | **54.36** |

Table 4: Comparison with state-of-the-art methods on the R-RefCOCO/+/g dataset.

| Method | Backbone | oIoU | mIoU | Acc. |
|---|---|---|---|---|
| *MLLM Methods* | | | | |
| LISA-V-7B (ft) | SAM-ViT-H | 65.39 | 66.41 | 93.39 |
| GSVA-V-7B (ft) | SAM-ViT-H | 68.13 | 68.29 | 94.59 |
| *Specialist Methods* | | | | |
| MCN | DarkNet-53 | 54.70 | 55.03 | 75.81 |
| VLT | DarkNet-53 | 60.43 | 60.21 | 79.26 |
| LAVT | Swin-B | 64.78 | 64.45 | 83.11 |
| DMMI | Swin-B | 68.21 | 68.77 | 87.02 |
| HDC | Swin-B | 69.31 | 68.81 | 93.34 |
| **IGVG (Ours)** | ViT-B | **71.52** | **71.12** | **97.42** |

Table 2: Comparison with state-of-the-art methods on the Ref-ZOM dataset.

| Methods | Val | | TestA | | TestB | |
|---|---|---|---|---|---|---|
| | F1score | N-acc. | F1score | N-acc. | F1score | N-acc. |
| MCN | 28.0 | 30.6 | 32.3 | 32.0 | 26.8 | 30.3 |
| VLT | 36.6 | 35.2 | 40.2 | 34.1 | 30.2 | 32.5 |
| MDETR | 42.7 | 36.3 | 50.0 | 34.5 | 36.5 | 31.0 |
| UNINEXT | 58.2 | 50.6 | 46.4 | 49.3 | 42.9 | 48.2 |
| SimVG | 62.1 | 54.7 | 64.6 | 57.2 | 54.8 | 57.2 |
| **IGVG** | **73.5** | **72.8** | **70.2** | **71.1** | **60.8** | **65.2** |

Table 3: GREC benchmark results on the gRefCOCO dataset. The threshold is set to 0.7 for all the methods.

## 4.2 MAIN RESULTS

**Results on GRES.** To evaluate the effectiveness of our approach in a generalized setting, we first conduct a comparative analysis with the existing specialized methods on the gRefCOCO dataset (Liu et al., 2023a), as presented in Tab. 1. The results demonstrate that our method establishes new state-of-the-art performance across all the metrics in three evaluation sets of the large-scale GRES benchmark. Notably, compared with the existing state-of-the-art method HDC (Luo et al., 2024), IGVG surpasses it with significant improvements of $+5.1\%$, $+2.7\%$, and $+2.9\%$ in gIoU on the val, testA, and testB sets, respectively. Furthermore, we report our results on the Ref-ZOM benchmark (Hu et al., 2023) in Tab. 2. Our method consistently outperforms the other methods under a fair comparison, achieving $+5.7\%$ improvement in Accuracy, $+2.4\%$ in oIoU, and $+2.7\%$ in mIoU. It is worth highlighting that our approach even surpasses GSVA (Xia et al., 2024), which leverages Multi-Modal Large Language Models (MLLM) (Liu et al., 2023c). In addition, we extend our evaluation to the R-RefCOCO/+/g datasets (Wu et al., 2024). As illustrated in Tab. 4, our method achieves substantial improvements of $+9.6\%$, $+10.0\%$, and $+10.5\%$ in rIoU for the R-RefCOCO/+/g tasks when compared to HDC.

| Multi-task | Ins.-aware | PIPH | F1score | N-acc. | gIoU | cIoU |
|------------|------------|------|---------|--------|------|------|
| | | | 65.98 | 66.13 | 65.03 | 64.77 |
| ✓ | | | 67.13 | 69.98 | 67.33 | 64.90 |
| ✓ | ✓ | | 68.42 | 71.86 | 70.13 | 65.88 |
| ✓ | ✓ | ✓ | **71.43** | **75.87** | **72.41** | **67.39** |

Table 5: Effectiveness of the core modules.

| STS | DSPS | ITQ | F1score | N-acc. | gIoU | cIoU |
|-----|------|-----|---------|--------|------|------|
| | | | 69.20 | 69.95 | 70.43 | 66.36 |
| ✓ | | | 69.09 | 70.80 | 70.51 | 66.16 |
| ✓ | ✓ | | 71.16 | 72.87 | 71.73 | **67.40** |
| ✓ | ✓ | ✓ | **71.43** | **75.87** | **72.41** | 67.39 |

Table 6: Effects of the AQG components.

| Query Decoder | F1score | N-acc. | gIoU | cIoU |
|---------------|---------|--------|------|------|
| DETR | 67.87 | 71.55 | 70.72 | 66.70 |
| Def. DETR | 69.14 | 72.31 | 71.23 | 66.87 |
| MS Def. DETR | 70.98 | 74.22 | 71.49 | 67.08 |
| Point-prior MS Def. DETR | **71.43** | **75.87** | **72.41** | **67.39** |

Table 7: Comparison of different decoders.

| Global Sup. | Ins. Sup. | Neg. Sup. | F1score | N-acc. | gIoU | cIoU |
|-------------|-----------|-----------|---------|--------|------|------|
| ✓ | | | 67.77 | 67.63 | 67.90 | 65.77 |
| | ✓ | | 69.46 | 72.96 | 69.19 | 65.15 |
| ✓ | ✓ | | **71.71** | 74.45 | 72.15 | 66.91 |
| ✓ | ✓ | ✓ | 71.43 | **75.87** | **72.41** | **67.39** |

Table 8: Impact of different level of supervision.

**Results on GREC.** In addition to performing general segmentation, our IGVG model is also capable of handling detection tasks. We evaluate the detection performance of IGVG on the GREC (He et al., 2023) dataset and compare it with existing state-of-the-art methods. The results are presented in Tab. 3. Notably, under the same threshold of 0.7, IGVG significantly outperforms the existing state-of-the-art method SimVG (Dai et al., 2024) with the improvements of $+11.4\%$, $+5.6\%$, and $+6.0\%$ in F1score on the validation, testA, and testB sets, respectively.

## 4.3 ABLATION STUDY

### 4.3.1 EFFECTIVENESS OF THE CORE MODULES

The core issues discussed in this paper include: 1) the impact of multi-task joint learning on generalized visual grounding; 2) the influence of fine-grained instance-aware perception; and 3) the effectiveness of the proposed Point-guided Instance-aware Perception Head. As shown in Tab. 5, multi-task joint supervision positively contributes to task complementarity in generalized scenarios, leading to performance improvements in both the GREC and GRES benchmarks. Specifically, the F1score is increased by 1.2% and gIoU is improved by 2.3%. After incorporating an instance-level segmentation branch to enable query-guided instance perception, the F1score is improved by 1.3% and gIoU is increased by 3.2%. Finally, introducing point priors to guide both instance and object predictions for all queries resulted in a further improvements of 3.0% in $F_1$score and 2.3% in gIoU.

### 4.3.2 THE POINT-GUIDED INSTANCE-AWARE PERCEPTION HEAD

**Analysis of Attention-guided Query Generation.** First, the Score Text Selector (STS) chooses $N_q$ highly responsive tokens from the $N_t$ tokens of $\mathcal{F}_t$, thereby reducing the computational cost of multi-head cross attention. As observed in Tab. 6, the introduction of STS results in almost no accuracy loss. The Dist-Score Point Selector (DSPS) selects $N_q$ prior reference points covering different instances based on attention responses. In contrast, the baseline uses a strategy of selecting the Top $N_q$ points, and DSPS improves F1score by 2.1% and gIoU by 1.2%. Lastly, we designed Inject Text Query (ITQ) to assist in learning the attention map. This not only helps to optimize the attention map but also injects text information into the initial query, resulting in a +3.0% improvement in N-acc and a +0.7% increase in gIoU.

**Analysis of Different Decoders.** As shown in Tab. 7, our baseline method uses the original DETR (Carion et al., 2020) decoder. By introducing the Deformable DETR decoder, we observe an improvement of 1.3% in F1score and 0.5% in gIoU. Furthermore, we design a multi-scale feature map using SimFPN, and by employing a hierarchical multi-scale Deformable DETR decoder, N-acc increases by 1.9%. Finally, using the points filtered by AQG as the initial reference points for the Deformable DETR yields a further improvement of 0.5% in F1score and 0.9% in gIoU.

**Analysis of Different Levels of Supervision.** As shown in Tab. 8, the impact of different levels of semantic supervision on performance is significant. Instance-level supervision alone outperforms global-level supervision, enhancing both detection and segmentation performance by equipping the model with instance awareness. Interestingly, we find that the joint training with both global and instance segmentation improves instance-aware performance even without fusion during post-processing. We hypothesize that this is due to global supervision encouraging $S_{\text{global}}$ to produce

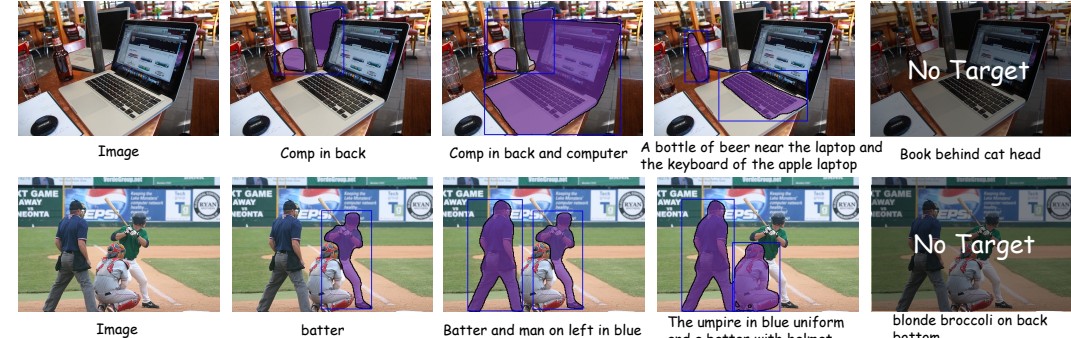

Figure 5: **Multi-task Visual Grounding Results.** Both GREC and GRES results for the same image under different expressions.

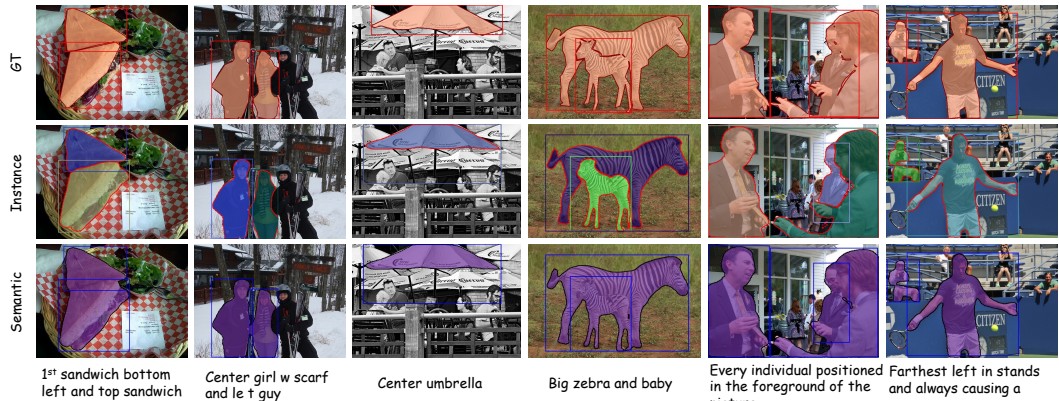

Figure 6: **Instance-level Segmentation Results.** The 'Instance' row presents instance-level segmentation. The 'Semantic' row presents the combination of both segmentation and instance masks.

strong semantic representations, thereby enhancing the discriminability of constructing the query mask $Q_s$. Finally, we introduce negative sample supervision, which guides the model to suppress mask predictions for negative sample queries.

### 4.4 QUALITATIVE RESULTS

IGVG effectively integrates and jointly accomplishes both the GREC and GRES tasks. Fig. 5 demonstrates the synchronized execution of detection and segmentation by IGVG, highlighting its ability to handle these tasks concurrently. Furthermore, IGVG exhibits instance-aware capabilities, enabling more fine-grained instance-level segmentation. Fig. 6 illustrates instance-level predictions, along with the combined final predictions. More visualization can be found in Appendix F.

## 5 CONCLUSION

This paper presents the Instance-aware Generalized Multi-task Visual Grounding (IGVG) framework, which, for the first time, unifies the GREC and GRES tasks while exploring the feasibility of instance-aware perception in GRES. Additionally, we propose a novel Point-guided Instance-aware Perception Head (PIPH) that adaptively selects prior reference points through attention maps, incorporating spatial priors into queries to enhance instance-specific targeting. Furthermore, by establishing associations between queries, objects, and instances, we achieve consistent predictions for points, boxes, and masks. Lastly, our IGVG framework significantly outperforms existing methods across gRefCOCO (GREC/GRES), Ref-ZOM, and R-RefCOCO/+/g datasets.

## 6 ETHICS STATEMENT

We acknowledge the ICLR Code of Ethics and affirm that our work adheres to its principles. Our research does not involve human subjects, nor does it raise concerns regarding discrimination, bias, or fairness. We have taken precautions to ensure data privacy and security and have complied with all relevant legal and ethical standards. No potential conflicts of interest or sponsorship influence the findings of this study.

## 7 REPRODUCIBILITY STATEMENT

We are committed to ensuring the reproducibility of our work. All key experimental details, including model architecture, hyperparameters, and training settings, are provided in the main text and appendix. Upon acceptance, we will release the source code and datasets used in this study to facilitate reproducibility.

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
