APPENDIX

## A DATASETS

**gRefCOCO.** This dataset comprises 278,232 expressions, including 80,022 referring to multiple targets and 32,202 to empty targets. It features 60,287 distinct instances across 19,994 images, which are divided into four subsets: training, validation, testA, and testB, following the UNC partition of RefCOCO (Yu et al., 2016).

**Ref-ZOM.** Ref-ZOM is derived from the COCO dataset (Lin et al., 2014), consisting of 55,078 images and 74,942 annotated objects. Of these, 43,749 images and 58,356 objects are used for training, while 11,329 images and 16,586 objects are designated for testing. Annotations cover three scenarios: one-to-zero, one-to-one, and one-to-many, corresponding to empty-target, single-target, and multiple-target cases in GRES, respectively.

**R-RefCOCO.** This dataset includes three variants: R-RefCOCO, R-RefCOCO+, and R-RefCOCOg, all based on the classic RES benchmark, RefCOCO+/g (Yu et al., 2016). Only the validation set adheres to the UNC partition principle, which is officially recognized for evaluation. The dataset formulation incorporates negative sentences into the training set at a 1:1 ratio with positive sentences.

## B METRICS

For GRES, we evaluate our model's performance using Pr@0.7, gIoU, cIoU, and N-acc metrics for gRefCOCO (Liu et al., 2023a). For Ref-ZOM, we adopt oIoU and mIoU metrics as defined in (Hu et al., 2023). R-RefCOCO (Wu et al., 2024) metrics include mIoU, mRR, and rIoU, all of which are specified in their respective benchmarks. The Generalized IoU (gIoU) calculates the average IoU for each image across all instances. In cases of empty targets, true positive IoU values are considered as 1, while false negatives are assigned 0. The cIoU metric evaluates the total intersection pixels relative to the total union pixels. In Ref-ZOM, mIoU represents the average IoU for all images containing referred objects, and oIoU is equivalent to cIoU. For R-RefCOCO, rIoU quantifies robust segmentation quality by factoring in negative sentences, assigning equal weight to positive instances in the mIoU calculation. N-acc. in gRefCOCO and Acc. in Ref-ZOM are defined similarly, representing the ratio of correctly classified empty-target expressions to the total empty-target expressions in the dataset. Additionally, mRR in R-RefCOCO computes the recognition rate for empty-target expressions per image and averages these across the dataset.

For GREC, we assess the percentage of samples achieving an F1score of 1 with an IoU threshold of 0.5. A predicted bounding box is classified as a true positive (TP) if it matches a ground-truth bounding box with an IoU of at least 0.5; if multiple predictions match, only the one with the highest IoU counts as TP. Ground-truth boxes without matches are false negatives (FN), while unmatched predicted boxes are false positives (FP). The F1score for a sample is computed as F1score $= \frac{2TP}{2TP+FN+FP}$, with samples deemed successfully predicted if their F1score is 1. For samples lacking targets, the F1score is 1 if no predictions exist, otherwise it is 0.

## C ADDITIONAL IMPLEMENTATION DETAILS

The maximum sentence length is limited to 50 words, and the images are resized to $320 \times 320$. We train our models for 10 epochs with a batch size of 16, utilizing the Adam optimizer (Kingma & Ba, 2014). All experiments are conducted on a system with dual NVIDIA 4090 GPUs, without employing the Exponential Moving Average (EMA) technique. The initial learning rate for the Multi-Modality Encoder (MME) is set to $5 \times 10^{-5}$, while other parameters are set at $5 \times 10^{-4}$. The learning rate decays by a factor of 0.1 at the 7th epoch to ensure comprehensive results. All ablation studies are performed at a resolution of $224 \times 224$, with training spanning 10 epochs and the same learning rate decay occurring at the 7th epoch. Metrics are based on the validation split of the gRefCOCO dataset. By default, the hyperparameters in Eq. 4 are set as follows: $\lambda_{\text{cls}} = 1.0$,

$\lambda_{\text{box}} = 5.0$, $\lambda_{\text{giou}} = 2.0$, and $\lambda_{\text{point}} = 2.0$. The weight parameters in Eq. 6 are set as: $\lambda_{\text{grec}} = 0.1$, $\lambda_{\text{global}} = 1.0$, $\lambda_{\text{instance}} = 1.0$, $\lambda_{\text{exist}} = 0.2$, and $\lambda_{\text{neg}} = 0.2$.

# D ADDITIONAL METHODS

## D.1 SCORE TEXT SELECTOR

---
**Algorithm 2** Score Text Selector

---
**Require:** Feature set $\mathbf{F} \in \mathbb{R}^{L \times C}$, mask $\mathbf{m} \in \{0, 1\}^L$, selection number $N$
**Ensure:** Selected feature set $\mathbf{F}_{\text{selected}} \in \mathbb{R}^{N \times C}$, selected mask $\mathbf{m}_{\text{selected}} \in \{0, 1\}^N$
 1: Mask and extract valid features: $\mathbf{F}_{\text{valid}} = \mathbf{F} \odot \mathbf{m}$
 2: Compute L2 norm scores for valid features: $\mathbf{s} = \|\mathbf{F}_{\text{valid}}\|_2$
 3: Count valid features: $V = \sum \mathbf{m}$
 4: **if** $V \geq N$ **then**
 5:     Select top-$N$ features based on scores: $\mathbf{F}_{\text{selected}} = \text{TopK}(\mathbf{F}_{\text{valid}}, N)$
 6:     Set selected mask: $\mathbf{m}_{\text{selected}} = \mathbf{1}^N$
 7: **else**
 8:     Select all valid features: $\mathbf{F}_{\text{selected}} = \mathbf{F}_{\text{valid}}$
 9:     Pad to $N$ features: $\mathbf{F}_{\text{selected}} \leftarrow \text{Pad}(\mathbf{F}_{\text{selected}}, N)$
10:     Set selected mask for valid features: $\mathbf{m}_{\text{selected}} = \text{Pad}(\mathbf{m}, N)$
11: **end if**
12: **return** $\mathbf{F}_{\text{selected}}, \mathbf{m}_{\text{selected}}$

---

The primary function of the Score Text Selector algorithm is to select a specified number of high-response features from a feature set based on a given mask. First, the algorithm filters the valid features using the mask and calculates their L2 norm scores. Then, it compares the number of valid features with the predefined selection number $N$. If the number of valid features is greater than or equal to $N$, the top $N$ features with the highest scores are selected, and the corresponding mask is set to all ones. Otherwise, all valid features are selected, and padding is applied to reach $N$ features, with the mask being filled accordingly. Finally, the algorithm returns the selected feature set and the corresponding mask.

## D.2 POST-PROCESS

Due to the introduction of instance-level segmentation masks, the post-processing of the GRES task differs significantly from previous GRES approaches. The pipeline is illustrated in Fig. 7. First, we weight the query scores and the non-target score to reduce false positives from single instances in scenes without targets. A threshold $thr_q$ is used to obtain the indices of valid queries, denoted as $index$. The detection

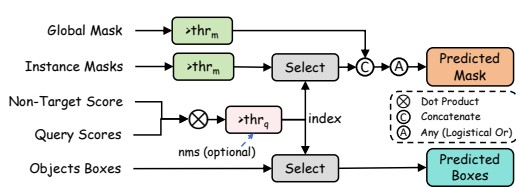

Figure 7: **Illustration of Post-processing**.

branch directly filters and outputs the corresponding targets based on these indices. The segmentation branch involves combining the global mask with instance masks. A threshold $thr_m$ is applied to select the pixel-level foreground mask. Then, the global mask is concatenated with the instance masks filtered by $index$, followed by a logical OR operation to address incomplete instances.

# E ADDITIONAL ABLATION STUDIES

| $\lambda_{point}$ | F1score | gIoU | cIoU |
|---|---|---|---|
| 1.0 | 70.37 | 71.53 | 66.95 |
| 2.0 | **71.43** | **72.41** | **67.39** |
| 5.0 | 69.90 | 71.47 | 66.85 |
| 10.0 | 69.28 | 97.31 | 66.82 |

Table 9: Impact of different ratios of point cost.

| $N_q$ | F1score | N-acc. | gIoU | cIoU |
|---|---|---|---|---|
| 3 | 70.26 | 72.74 | 71.32 | 66.74 |
| 5 | **71.60** | 73.85 | 71.55 | 66.87 |
| 10 | 71.43 | **75.87** | **72.41** | **67.39** |
| 20 | 69.16 | 73.29 | 71.57 | 66.95 |
| 30 | 68.55 | 71.90 | 71.22 | 66.63 |

Table 10: Impact of number of queries.

| Mask Output | gIoU | cIoU |
|---|---|---|
| Only Global | 72.61 | 65.66 |
| Only Instance | 74.19 | 67.18 |
| Merge | **74.65** | **67.66** |

Table 11: Impact of mask output in post-processing.

| Non-Tar. Weighted | NMS | F1score | gIoU | cIoU |
|---|---|---|---|---|
| | | 73.18 | 73.94 | 67.20 |
| ✓ | | 74.38 | **74.59** | **67.58** |
| ✓ | ✓ | **74.71** | 74.55 | 67.48 |

Table 12: Impact of non-target weighting and NMS in post-processing.

### E.1 THE EFFECT OF POINT COST WEIGHT

In the Point-guided Target Matcher, we introduce an additional point cost to the original DETR cost function. We conducted ablation studies to assess the impact of the point cost weight $\lambda_{point}$, as shown in Tab. 9. From the experimental results, we select $\lambda_{point} = 2$.

### E.2 THE IMPACT OF THE NUMBER OF POINTS

Since IGVG establishes a one-to-one correspondence between queries and reference points, their quantities must match. We conducted experiments to explore the effect of the number of reference points on performance. As shown in Tab. 10, increasing $N_q$ generally requires longer training times to achieve convergence. After balancing these considerations, we select $N_q = 10$.

### E.3 THE IMPACT OF POST-PROCESS

**The impact of mask merge.** IGVG generates both global and instance-level segmentations. We analyzed the performance of these predictions both individually and when combined, as shown in Tab. 11. The instance-level predictions, which benefit from finer-grained supervision, achieve better performance compared to global predictions, improving gIoU by +1.6%. Furthermore, merging the global and instance-level predictions yields an additional 0.5% improvement in gIoU.

**The impact of NT score and NMS.** As demonstrated in Tab. 12, we evaluated the effects of integrating the Non-Target (NT) branch's score into the query score and the influence of Non-Maximum Suppression (NMS). The introduction of the NT score effectively incorporates global confidence into each instance, resulting in a +1.2% F1score and +0.7% gIoU.

## F ADDITIONAL VISUALIZATION

In Fig. 8, we provide additional visualizations of IGVG's intermediate processes, including the points corresponding to the queries, the predicted boxes, and masks. It can be observed that IGVG achieves consistency across points, boxes, and masks for individual instances. Additionally, we visualize the attention maps from the Attention-based Query Generation Module and the corresponding selected points. In Fig. 9, we present examples of multi-object scenarios from the Ref-ZOM dataset. While IGVG can perceive object locations, we find that its detection accuracy for small objects remains insufficient, mainly due to the limitations imposed by the model's input size. In Fig. 10, we visualize the results of the three subsets of the R-RefCOCO dataset: R-RefCOCO, R-RefCOCO+, and R-RefCOCOg.

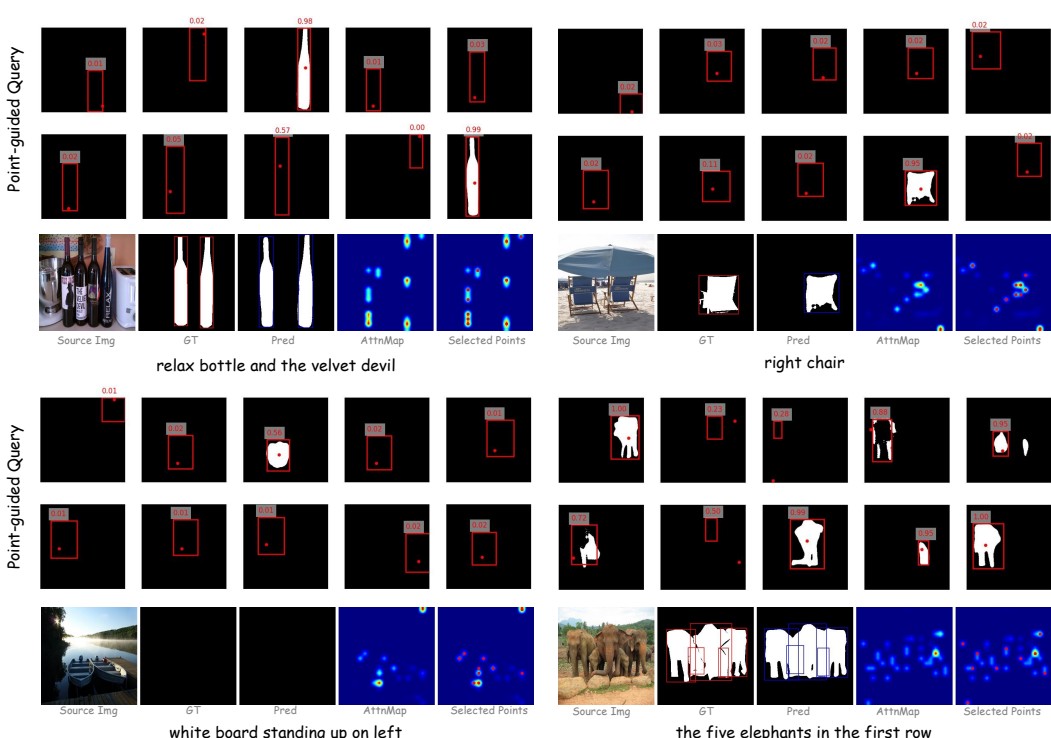

Figure 8: **Visualization of IGVG Details.** The "Point-guided Query" illustrates the points corresponding to each query, along with the predicted bounding boxes and masks. "AttnMap" represents the Attention Map from the Attention-based Query Generation module, while "Selected Points" indicates the reference points output by the Dist-Score Point Selector.

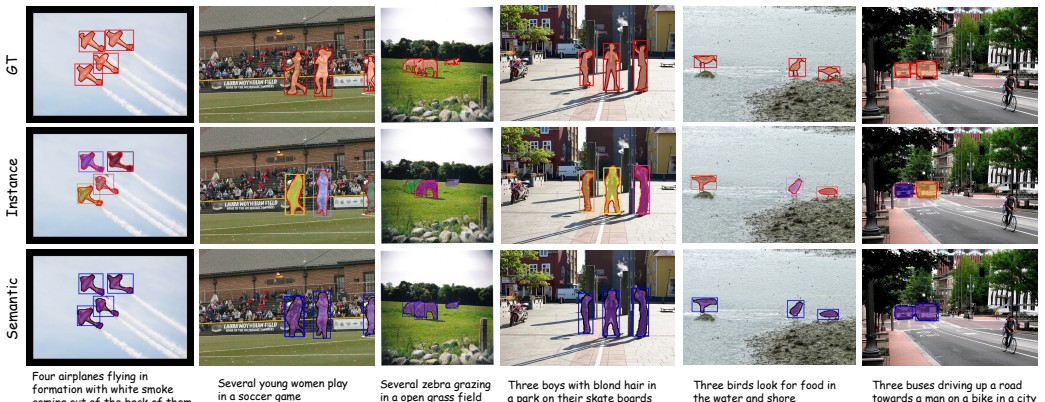

Figure 9: **Visualization of multi-object situations in the Ref-ZOM dataset.**

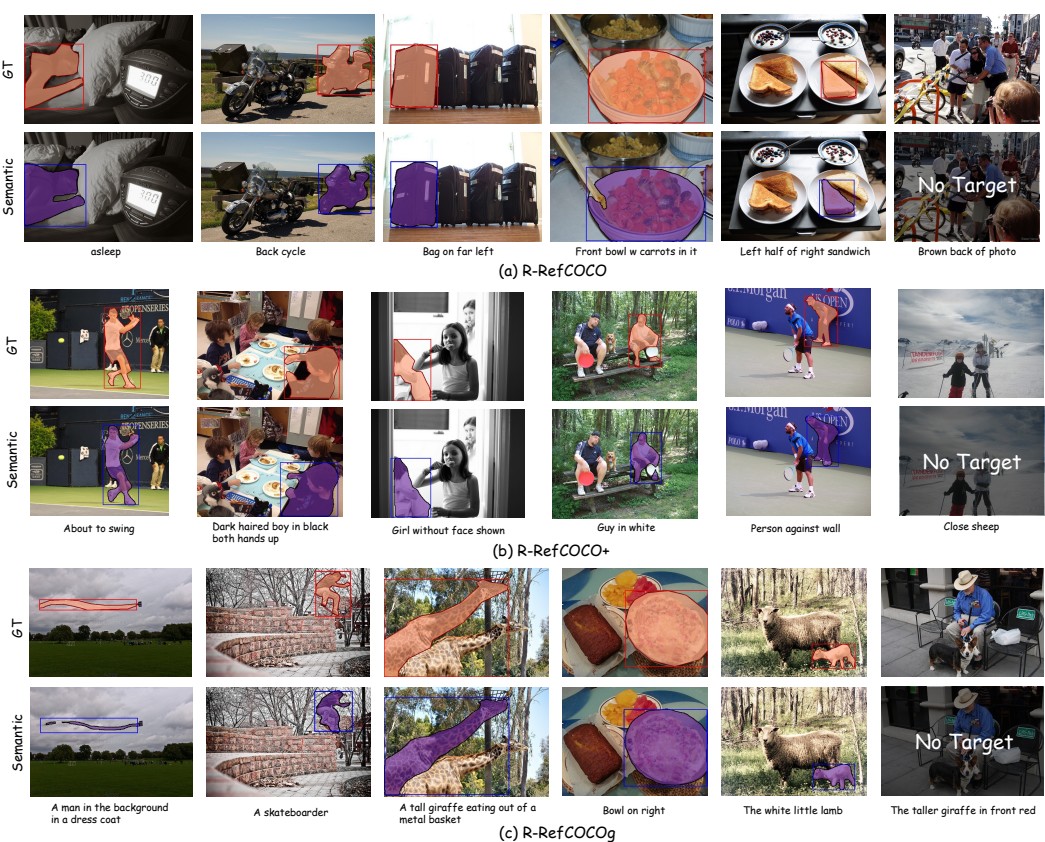

Figure 10: **Visualization of R-RefCOCO dataset.**