# OpenReview forum: "Instance-aware Generalized Multi-task Visual Grounding"
_ICLR.cc/2025/Conference — ICLR 2025 Conference Withdrawn Submission_

### Official Review · Reviewer_dYUB · 2024-10-25

**Soundness:** 3
**Presentation:** 1
**Contribution:** 2
**Rating:** 5
**Confidence:** 5

**Summary:**

The paper proposes a general framework for GRES and GREC task called IGVG. IGVG presents a Point-guided Instance-aware Perception Head (PIPH), which employs attention-based query generation to identify coarse reference points for better results.

**Strengths:**

1. Experiments are sufficient to prove the effect of the IGVG, and the improvement is impressive.

2. The motivation is clear to utilize the multi-task to improve the performance.

**Weaknesses:**

1. Writing. The writing and structure is poor. The paper has a lot of unclear claim. Like What is the 'D' in the 8-th line of the Algorithm 1? What is the 'ITQ' in Tab. 6? It seems like a part of AQG but never be mentioned before. At the same time, why move some content to supplementary materials while there is space left? The appendix is like HDC [1].

2. Contribution. The key components of IGVG are from many existing methods like deformable-detr [2], SimFPN [3], and Unet. The proposed PIPH takes multiply points to assist the process of segmentation, it has been explored in PPMN [4] and NICE [5] to solve the multi-object RES and REC. So the contribution is limited.


[1] HDC: Hierarchical Semantic Decoding with Counting Assistance for Generalized Referring Expression Segmentation

[2] Deformable DETR: Deformable Transformers for End-to-End Object Detection

[3] Exploring plain vision transformer back- bones for object detection.

[4] PPMN: Pixel-Phrase Matching Network for One-Stage Panoptic Narrative Grounding

[5] NICE: Improving Panoptic Narrative Detection and Segmentation with Cascading Collaborative Learning \\

**Questions:**

1. IGVG takes a MME as the feature extractor. Is the MME pre-trained to align the language and vision space?

2. According the post-process, the final mask relies on the global mask and the instance masks, and the instance masks are after the detection result. So how to solve the conflict between the detection branch and the AvgPool of the global mask? Like the global mask predicts no target but the detection branch returns boxes.

3. In the implementation details, why the resolution is different between ablation study and the main results?

4. The number of points in PIPH is fixed, so how to solve the point prediction errors caused by dense targets and the points that are also on the target but predicted as no target?

---

### Official Review · Reviewer_gLyn · 2024-11-03

**Soundness:** 2
**Presentation:** 2
**Contribution:** 3
**Rating:** 5
**Confidence:** 3

**Summary:**

This paper presents a new Instance-aware Generalized multi-task Visual Grounding (IGVG) framework that combines the Referring Expression Segmentation (GRES) task and Comprehension (GREC) tasks.

**Strengths:**

1. The motivation behind this work is evident and compelling.
2. The performance of this method is state-of-the-art.

**Weaknesses:**

1. Where is the Point-guided Instance-aware Perception Head (PIPH) in Figure 2? The description of Figure 2 is quite complex, making it challenging to grasp the central idea of this paper.
2. The writing needs improvement, specifically in Section III where the crucial method descriptions are unclear.

**Questions:**

see above mentioned.

---

### Official Review · Reviewer_3xJD · 2024-11-04

**Soundness:** 2
**Presentation:** 3
**Contribution:** 2
**Rating:** 5
**Confidence:** 4

**Summary:**

This paper presents the instance-aware generalized multi-task visual grounding framework, which unifies the GREC and GRES tasks while exploring the feasibility of instance-aware perception in GRES. They propose a point-guided Instance-aware perception head that adaptively selects prior reference points through attention maps, incorporating spatial priors into queries to enhance instance-specific targeting.

**Strengths:**

- The motivation for unifying GRES and GREC tasks presented in this paper makes sense;

- The experiment in this paper is relatively sufficient;

**Weaknesses:**

---

Q1. In the visual grounding task, the concept of "instance-level perception" proposed in this paper is not novel, and there have been numerous similar studies proposing related approaches. The emphasis on "instance-level" in this paper lacks enough innovation.

Q2. The highlighted aspect of "point-guided instance-aware perception" in this paper is also not novel, as several other works like Ferret V1 [1] and Dynamic-MDETR [2] have proposed similar ideas. However, this paper does not delve deeper discussion into it.

Q3. This paper lacks thorough literature research since it fails to cite or discuss more various multi-task grounding studies, such as VG-LAW [3], OneRef [4], UniQRNet [5], etc., where OneRef also employ BEiT-3 as their backbone structure.

Q4. This paper appears to be an incremental work built upon SimVG [6].

Q5. The supplementary materials should include the results of both single-task and multi-task experiments conducted using the proposed method on the classical RefCOCO/+/g dataset.

Q6. The paper does not clearly explain the differences between gRefCOCO, R-RefCOCO, and classical RefCOCO, nor does it explain in the supplementary materials which literature these datasets come from (so as Ref-ZOM dataset).

Q7. (Writing issue) This paper defines various professional terms and introduces a lot of strange custom abbreviated nouns (such as REP, TP, IP, STS, MME, AQG, DSPS, PIPH), such writing is very irregular and problematic. For example: (1) "text projection" is abbreviated as TP, while in the later of the paper, it is clarified that TP stands for "True positive"; (2) The BEiT-3 paper does not refer to itself as "MME" while this paper named BEiT-3 backbone as "MME".

Q8. (Format issue) The references in this paper are not hyperlinked resulting in a poor review experience.

--

[1] Ferret: Refer and ground anything anywhere at any granularity. ICLR 2023.

[2] Dynamic mdetr: A dynamic multimodal transformer decoder for visual grounding[J]. TPAMI 2023.

[3] VG-LAW: Language adaptive weight generation for multi-task visual grounding. CVPR 2023.

[4] OneRef: Unified One-tower Expression Grounding and Segmentation with Mask Referring Modeling. NeurIPS 2024.

[5] Uniqrnet: Unifying referring expression grounding and segmentation with qrnet. TOMM 2024.

[6] SimVG: A Simple Framework for Visual Grounding with Decoupled Multi-modal Fusion. 2024.

**Questions:**

Please see weakness.

---

### Note · Authors · 2024-11-26

I have read and agree with the venue's withdrawal policy on behalf of myself and my co-authors.